# Trajectories of Changes in Phytoplankton Biomass, *Phaeocystis globosa* and Diatom (incl. *Pseudo-nitzschia sp.*) Abundances Related to Nutrient Pressures in the Eastern English Channel, Southern North Sea

**Alain Lefebvre \***  **and Camille Dezécache**

IFREMER (French Research Institute for Exploitation of the Sea), Laboratoire Environnement Ressources, 62321 Boulogne-sur-Mer, France; Camille.Dezecache@ifremer.fr
\*   Correspondence: Alain.Lefebvre@ifremer.fr

**Abstract:** The phytoplankton compartment is particularly reactive to changes in nutrient concentration and is used as a quality indicator. Using a simple numerical approach, the response of emblematic harmful taxa from the eastern English Channel and southern North Sea to changes in nutrient inputs was studied. The method is based on a diachronic approach using averaged maxima over sliding periods of six years (1994–2018). This gave a final dataset containing pairs of points (number of years) for explained and explanatory variables. The temporal trajectory of the relationship between each pair of variables was then highlighted. Changes were represented as long-term trajectories that allowed a comparison to a reference/average situation. In addition, the relevance of the use of *Phaeocystis globosa* and the *Pseudo-nitzchia* complex as eutrophication species indicators was tested. Results showed a significant shift in the 2000s and different trajectories between diatoms and *P. globosa* abundances in response to changes in Dissolved Inorganic Nitrogen (DIN). The contrasting ecosystems under study reacted differently depending on the initial pressure. While a return to good ecological status does not seem feasible in the short term, it seems that these ecosystems were in an unstable intermediate state requiring continued efforts to reduce nutrient inputs.

**Keywords:** harmful algal blooms; eutrophication; trajectory; *Phaeocystis globosa*; *Pseudo-nitzchia* complex; Oslo and Paris Convention OSPAR; Water Framework Directive (WFD); Marine Strategy Framework Directive (MSFD)

## 1. Introduction

The importance of phytoplankton at the level of aquatic ecosystems is no longer in doubt (base of the food web, key component of the structure and function of coastal marine ecosystems and carbon and nutrient budgets, etc.). Ocean primary productivity is largely determined by phytoplankton growth, which can be affected by climate change as well as direct human activities through eutrophication. Coastal areas concentrate human activities benefiting from incomes deriving from sea uses, as well as areas of high phytoplankton concentrations leading to environmental disturbances or eventual toxicity of sea products. Indeed, among the thousands of taxa contributing to this phytoplankton community, a few are classified as toxic or harmful, and are referred to as Harmful Algal Bloom (HAB).

HAB are proliferations of phytoplankton (mostly dinoflagellates, diatoms and cyanobacteria) that have negative impacts on environments and associated biota (water discoloration, foam accumulation, toxin production leading to seafood contamination, mortality). In recent decades, HAB has increased in frequency and intensity, partly caused by eutrophication and warming [1,2]. Consequently, human society is affected with impacts on food provisioning, tourism, economy and human health.

For French coastal waters, *Alexandrium sp.*, *Dinophysis sp.* and *Pseudo-nitzschia sp.* are the most common HAB-toxin-producing species [3]. In the eastern English Channel and southern bight of the North Sea, *Phaeocystis globosa* is also classified as HAB. *P. globosa* blooms have deleterious effects on benthic and pelagic ecosystems; they cause deep ecosystem reorganization, negative effects on fisheries and aquaculture, and negative perception of the environment by tourists [4–9]. Moreover, *P. globosa* co-occurs with *P. delicatissima* complex [10–12] which may serve as a solid substrate during the transitional phase of its life stage [5,13]. In the studied area, three species of *Pseudo-nitzschia* were identified (*P. delicatissima, P. pungens, P. fraudulenta*) [14].

The great complexity of coastal and estuarine ecosystems driven by multifactorial non-linear interactions makes it particularly difficult to study the consequences of the disturbances they are facing [15]. For example, the increase in global estuarine chlorophyll observed and explained by climate change [16] may be easily masked by the natural variability of natural ecosystems or eutrophication. While it is always desirable to obtain the most precise information about species composition and hydrological parameters with the highest possible sampling frequency, these datasets still lack sufficient historical coverage to assess the environmental consequences of climate change or human activities, such as eutrophication, contrary to classic low-frequency in-situ monitoring. In addition to their long historical coverage, these low-frequency sampling techniques are often applied over contrasted ecosystems. This offers a great opportunity to apply both a diachronic approach, following changes in parameters over time in a particular site, and a synchronic approach, where spatial differences in the measured parameters can substitute temporal changes, and help to identify possible future trends.

Such a scenarization is of major importance for decision makers wishing to establish indicators of water quality and improvement objectives. In order to assess water quality and its changes in time, several biological metrics are used within European directives and regional sea conventions. Chlorophyll-*a* (Chl-*a*) is the most widely used indicator, and is considered as a proxy for total phytoplankton biomass. Biodiversity indicators are still in development and the existing ones are considered as surveillance indicators (without assessment level or threshold) [17,18]. In the meantime, it is necessary to monitor the evolution of the quality of the environment, so some simplifications are necessary. According to Tett et al. [19], "although there are no species that could serve as universal indicators of nutrient-induced disturbance, there are some species that may serve as indicator of disturbance in particular water types". There are debates about the relevance of *P. globosa* as an indicator for eutrophication, as low evidence was found in the literature between nutrients and *Phaeocystis* concentrations [20], with the possible development of *P. globosa* blooms under pristine conditions [21]. Moreover, the co-occurrence and potential competition for nutrients between *Phaeocystis* and diatoms may make eventual relationships difficult to evidence. *Phaeocystis* blooms chronologically, and follows diatom blooms, mainly controlled by silica concentrations [4,22]. Thus, a competition may be expected, in the sense that a depletion of nutrients by earlier diatom blooms may limit further expansion of *Phaeocystis*. Unclear conclusions published in the literature led to the advice that *Phaeocystis* should be excluded as indicator for eutrophication [20]. More unexpectedly, observations sometimes lead to an absence of relationship between nutrients and Chl-*a* concentrations [23].

While a lot of studies only focus on degradation trajectories of coastal ecosystems subjected to eutrophication processes [24,25], a few are able to consider both degradation and recovery because of limited data set [26–29]. The methods used can be more or less complex, therefore leading to a "wait and see" strategy from the stakeholders' side, under the pretext of uncertainty in scientific results. This kind of strategy pushes back the deadlines for improving the ecological status of marine waters. This is why, sometimes, we should go back to basic methodology in order to deliver clear, understandable results, and then associated recommendations for environmental management purposes.

The purpose of this study was thus to clarify the links between some biological indicators of eutrophication using a time series covering the period 1994–2018, with samples collected in contrasted environmental areas (estuary, coastal waters within a region of fresh water influence) covering different gradient of anthropogenic pressures. Relationships between Dissolved Inorganic Nitrogen (DIN)

concentrations and Chl-*a*, *Phaeocystis globosa* (later called *Phaeocystis* for simplicity matters) and diatom concentrations, respectively, were assessed. Changes in the Dissolved Silica/ Dissolved Inorganic Nitrogen (DSi/DIN) ratio were also followed, as this may influence the competitive advantage of diatoms compared to other taxonomical groups. Dissolved Inorganic Phosphorus (DIP) was not assessed here in detail as there is quite a consensus that efforts made to decrease its concentrations over recent decades have led to decreased eutrophication and no more significant reductions are expected. Preliminary results based on the same dataset indicated unclear relationships between the parameters involved and DIP concentrations, probably because any further decrease would not make much change to ecosystems functioning given the high DIN concentrations observed. Within the diatoms group, a secondary focus was made on *Pseudo-nitzschia*, a species known for occasionally provoking paralytic shellfish poisoning [12] and being sometimes concomitant to *Phaeocystis* blooms [30]. One major assumption of the current study, compared to previously published literature, was that the absence of relationships sometimes observed between variables could be the consequence of a high inter-annual variability and a quasi-systematic bias in measurements done compared to the theoretical values which were to be estimated. Indeed, the high variability of the parameters at a very short temporal scale, as observed with high-frequency measurement tools, indicates that it is very likely that peaks for nutrients or phytoplankton biomass are underestimated [31]. More precisely, in the best-case scenario, the sample would be taken at the precise moment of the peak, but more likely it measures a value corresponding to lower intensities characterizing pre-peak or post-peak situations (a theoretical example is given in Appendix A, Figure A1). To decrease the sensibility of the models to these quasi-systematic measurement biases, as well as to inter-annual variability of the parameters of interest, relationships were assessed using an averaged value along six-year sliding periods. This provides an opportunity to observe clearer relationships without underestimating the complexity of ecosystem functioning.

Our working hypotheses are as follows: (i) Reducing nutrients' riverine inputs has a rapid homogeneous effect on all studied ecosystems leading to lowering phytoplankton biomass. (ii) Resulting patterns of change in the phytoplankton community structure favored non-HAB species, (iii) The use of phytoplankton indicator species is relatively easy to understand where the impacts of these species may be observed (color changes, foaming, noxious odors). However, the growth of indicator species has not been shown to be primarily responsive to human activities. Our objective is to confirm or deny sensitivity of *Phaeocystis* to the management of nutrient inputs, or linkage in space and time to the availability of anthropogenic nutrients.

## 2. Materials and Methods

Data used derived from the Ifremer' SRN (Regional Monitoring on Nutrients) REPHY (Phytoplankton and Phycotoxins Monitoring Program) and IR ILICO PHYTOBS (Shoreline and Coastal Research Infrastructure Phytoplankton Observation) [32,33]. Three sampling areas are investigated, each being formed by an inshore–offshore transect (Figure 1). Two of them are located in the eastern English Channel: (i) Off Boulogne-sur-mer (BL1, BL2, BL3), a coastal zone separated from the open sea by a frontal area; (ii) in the Bay of Somme (S1, S2, S3), the second largest estuarine system after the Bay of Seine on the French coasts of the English Channel. The third study site is located off Dunkerque Harbour (DK1, DK3, DK4), a shallow well-mixed coastal area in the Southern Bight of the North Sea.

Chl-*a*, nutrients (DIN and DSi) and phytoplankton abundance were available bimonthly during the period from March to June, and monthly during the rest of the year for 1994–2018. Water samples were collected from sub-surface waters using a 5-L Niskin bottle [30]. Chl-*a* concentrations were estimated by spectrophotometry (Shimadzu UV-1700) after filtration through Whatman 47-mm GF/F glass fiber filters and extraction in acetone at 90% [34]. An acid Lugol's solution was used to preserve Phytoplankton samples collected from the Niskin bottle. 10-mL subsamples were settled for 24 h in a counting chamber following the Utermöhl method [35]. Cell enumeration was performed by inverted microscopy within 1 month of sample collection to avoid any significant changes in phytoplankton

size and abundance. Except for *Phaeocystis globosa*, over 400 phytoplankton cells in each sample were counted using a 20 × Plan Ph1 0.5NA objective, resulting in an error of 10%. For *Phaeocystis* enumeration, only the total number of cells was counted. A minimum of 50 solitary cells were enumerated from several randomly chosen fields (10 to 30) with a 40 × Plan Ph2 0.75NA. Cell abundance within a colony was estimated using a relationship between colony bio-volume and cell number proposed in the literature [8].

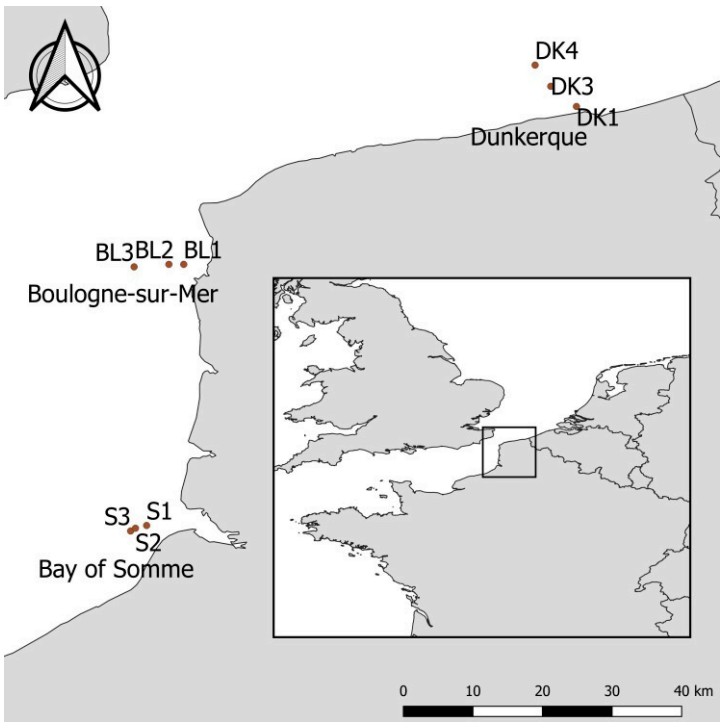

**Figure 1.** SRN network sampling stations located along the French coast in the eastern English Channel (Bay of Somme [S1, S2, S3] and Boulogne-sur-mer [BL1, BL2, BL3]) and in the Southern Bight of the North Sea (Dunkerque [DK1, DK3, DK4]). For each site, three sampling stations are sampled following a coastal–offshore gradient.

To align the two time series for hydrological parameters and phytoplankton samples, each sample was associated with its closest date from a regular reference time sequence beginning at 1 January 1994 and ending at December 31st, 2018.

Yearly maxima were calculated for all parameters, during the winter season (November–February) for nutrients and during the growing season (March–October) for biological parameters, as defined by the Water Framework Directive (WFD) [36]. The latter were used as proxies for the maximal accumulation of nutrients which can be consumed during the growing season to produce potential phytoplankton blooms of interest. However, due to their high short-term variability, averaging those maxima over long time periods is necessary to consistently estimate potential relationships between the parameters involved. Indeed, samples collected through the years are likely to miss the exact moment of nutrients or phytoplankton peaks, adding noise to the observed relationships (Figure A1 in Appendix A). This allows the emergence of consistent long-term trends from noisy data concerning the parameters considered.

Averaging values decreased the number of input data used for further statistical analysis, with the risk of characterizing a gradient of ecosystems, rather than a direct relationship between two variables, which is a common drawback of synchronic methods compared to diachronic approaches. To overcome this problem, averaged maxima per site over sliding periods of six years were calculated, corresponding to the duration of the assessment periods used in the WFD directive. This gave a final

dataset containing twenty pairs of points (number of years considered) for explained and explanatory variables. The temporal trajectory of the relationship between each pair of explained and explanatory variables were then observed, corresponding to the displacement of these pairs of points, along the different defined sliding periods. The consistency between each individual trajectory for the study sites and the overall trajectory for the whole period 1994–2018 was considered as sufficient proof that a "true" relationship was observed (Figure 2).

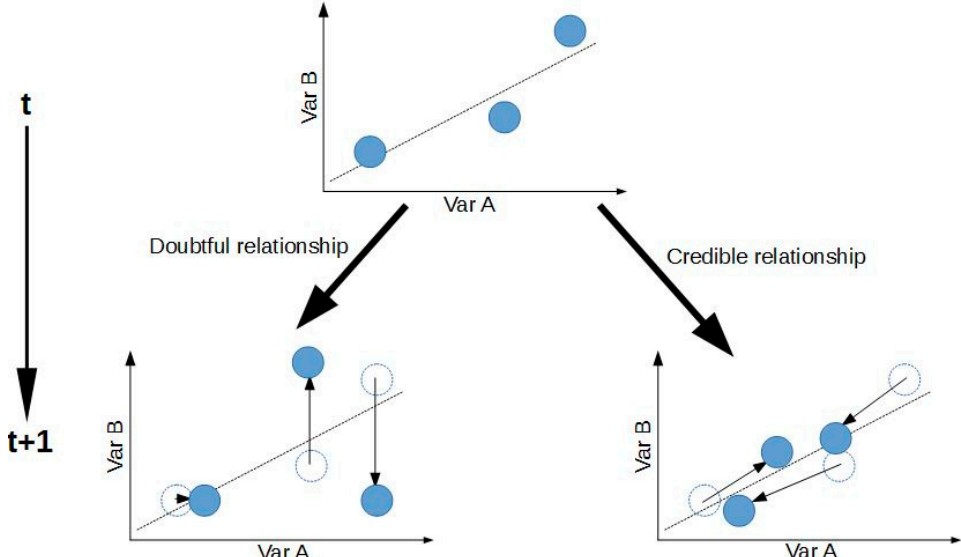

**Figure 2.** Credible vs. doubtful relationships between variables A and B in time. At time t, samples are illustrated by blue circles, each corresponding to a determined sampling site. At time t+1, they change to the direction indicated by the arrow. The associated linear regression for each date t is represented by a black dotted line, assumed to be stable over time. On the contrary, the individual trajectories of each sample are very different between both scenarios (left vs. right). In the first case (left), the trajectory for each site strongly differs from the linear regression, while in the second case (right), each trajectory closely follows the regression line's trend, whether increasing or decreasing, suggesting a consistent relationship between both variables.

All dataset processing and further statistical treatments were done using R software V4.0.0 [37] and the associated graphical package *ggplot2* [38]. The package *lubridate* [39] was used to process date and time information. Global trends were assessed using the R package *TTAinterfaceTrendAnalysis* [40]. Trend analyses were run on raw data from 1994 to 2018, without removal of outliers and data completion. To build a regularized time series, the interface aggregates raw data from the same period (month) using maximum or median methods. Because of its short bloom duration, trend analyses on *Phaeocytis globosa* data were done only for April–May from 1994 to 2018.

## 3. Results

### 3.1. Environmental Characteristics, Seasonnality and Trends

Although seasonal cycles were well defined on all transects, there were no differences in timing along coastal–offshore gradients (data not shown) [22,32,33]. Coastal–offshore gradients were classical with only differences in amplitude between stations from a given transect. Consequently, only the results from coastal stations are presented in Figures 3–5. Main statistical characteristics of environmental parameters and global trend results are synthetized in Tables 1 and 2.

Significant linear increasing and decreasing trends are observed, respectively, for Dinoflagellates, *Pseudo-nitzschia* complex and for DIN and DIP for all stations. Patterns of trends are different (even not significant) between stations for the other variables.

**Table 1.** Descriptive statistics on raw data from selected coastal stations (DK1, BL1 and S1) and parameters for the period 1994–2018. Minimum, mean, median, maximum, first (Q1) and third (Q3) quantiles, number of missing data (NA) and length (N) of each data series. *Phaeocystis*: statistics only for April–May.

| DK1 | Min | Q1 | Median | Mean | Q3 | Max | NA | N |
|---|---|---|---|---|---|---|---|---|
| **Chl-*a* (µg.L$^{-1}$)** | 0.24 | 2.55 | 4.55 | 6.95 | 8.60 | 53.18 | 6 | 321 |
| **DIN (µmol.L$^{-1}$)** | 0.30 | 1.51 | 5.35 | 10.47 | 17.65 | 54.92 | 17 | 310 |
| **DIP (µmol.L$^{-1}$)** | 0.01 | 0.17 | 0.40 | 0.51 | 0.69 | 4.90 | 15 | 312 |
| **DSi (µmol.L$^{-1}$)** | 0.1 | 1.05 | 3.13 | 5.05 | 6.57 | 35.20 | 14 | 313 |
| **Diatoms (cell.L$^{-1}$)** | 900 | 79,470 | 202,600 | 335,000 | 426,400 | 5,365,000 | 19 | 308 |
| **Dinoflagellates (cell.L$^{-1}$)** | 0 | 675 | 4685 | 11890 | 13,520 | 202,200 | 19 | 308 |
| ***Phaeocystis* (cell.L$^{-1}$)** | 0 | 0 | 465,800 | 4,061,000 | 6,473,000 | 28,230,000 | 2 | 78 |
| ***Pseudo-nitzchia* (cell.L$^{-1}$)** | 0 | 400 | 7016 | 51,600 | 34,650 | 1,505,000 | 19 | 308 |
| **BL1** | Min | Q1 | Median | Mean | Q3 | Max | NA | N |
| **Chl-*a* (µg.L$^{-1}$)** | 0.04 | 2.01 | 3.67 | 5.52 | 7.44 | 29.60 | 5 | 399 |
| **DIN (µmol.L$^{-1}$)** | 0.30 | 1.39 | 4.25 | 8.40 | 12.90 | 47.51 | 25 | 379 |
| **DIP (µmol.L$^{-1}$)** | 0.05 | 0.13 | 0.28 | 0.40 | 0.60 | 1.70 | 18 | 386 |
| **DSi (µmol.L$^{-1}$)** | 0.10 | 0.47 | 1.64 | 3.25 | 4.32 | 19.01 | 18 | 386 |
| **Diatoms (cell.L$^{-1}$)** | 400 | 88,740 | 210,800 | 374,300 | 417,000 | 7,492,000 | 5 | 399 |
| **Dinoflagellates (cell.L$^{-1}$)** | 0 | 1020 | 5362 | 10,450 | 12,600 | 237,500 | 5 | 399 |
| ***Phaeocystis* (cell.L$^{-1}$)** | 0 | 36,270 | 1,330,000 | 3,499,000 | 6,043,000 | 17,800,000 | 1 | 89 |
| ***Pseudo-nitzchia* (cell.L$^{-1}$)** | 0 | 400 | 8536 | 73,310 | 39,880 | 3,361,000 | 5 | 399 |
| **S1** | Min | Q1 | Median | Mean | Q3 | Max | NA | N |
| **Chl-*a* (µg.L$^{-1}$)** | 0.21 | 3.10 | 5.77 | 8.41 | 10.81 | 58.53 | 7 | 362 |
| **DIN (µmol.L$^{-1}$)** | 0.30 | 3.01 | 10.37 | 14.44 | 22.78 | 60.04 | 15 | 354 |
| **DIP (µmol.L$^{-1}$)** | 0.04 | 0.13 | 0.27 | 0.40 | 0.59 | 1.55 | 14 | 355 |
| **DSi (µmol.L$^{-1}$)** | 0.10 | 1.13 | 3.67 | 6.30 | 10.18 | 41.00 | 15 | 354 |
| **Diatoms (cell.L$^{-1}$)** | 2900 | 122,100 | 295,700 | 679,800 | 746,000 | 11,070,000 | 4 | 365 |
| **Dinoflagellates (cell.L$^{-1}$)** | 0 | 400 | 2900 | 12,460 | 10,920 | 815,100 | 4 | 365 |
| ***Phaeocystis* (cell.L$^{-1}$)** | 0 | 18,210 | 1,913,000 | 5,777,000 | 6,360,000 | 48,930,000 | 0 | 83 |
| ***Pseudo-nitzchia* (cell.L$^{-1}$)** | 0 | 0 | 3750 | 123,400 | 33,070 | 7,503,000 | 4 | 365 |

**Table 2.** Statistics of global trend analysis for the coastal stations D1, B1 and S1 for the period 1998–2018. Sen.slope: trend in original units per year; Sen.slope.pct: trend in percent of mean quantity per year, *p*-value: in bold when significant ($\alpha = 0.05$ or less).

| Parameter | Trend Statistics | DK1 | BL1 | S1 |
|---|---|---|---|---|
| **Chl-*a* (µg.L$^{-1}$)** | sen.slope | −0.055 | −0.0152 | 0.0235 |
| | sen.slope.pct | −0.0138 | −0.0063 | 0.0052 |
| | *p*-value | 0.005 | 0.2953 | 0.3774 |
| **DIN (µmol.L$^{-1}$)** | sen.slope | −0.173 | −0.0595 | −0.1272 |
| | sen.slope.pct | −0.028 | −0.0152 | −0.0157 |
| | *p*-value | 0 | $1.00 \times 10^{-4}$ | 0.0021 |
| **DIP (µmol.L$^{-1}$)** | sen.slope | −0.01 | −0.0087 | −0.004 |
| | sen.slope.pct | −0.0242 | −0.0247 | −0.0128 |
| | *p*-value | 0 | 0 | 0.0149 |

**Table 2.** *Cont.*

| Parameter | Trend Statistics | DK1 | BL1 | S1 |
|---|---|---|---|---|
| DSi (μmol.L$^{-1}$) | sen.slope | −0.0162 | 0.0278 | −0.0554 |
| | sen.slope.pct | −0.0063 | 0.024 | −0.0144 |
| | *p*-value | 0.3093 | 0.0017 | 0.0158 |
| Diatoms (cell.L$^{-1}$) | sen.slope | 2313 | 3411 | 2008 |
| | sen.slope.pct | 0.032 | 0.0376 | 0.0117 |
| | *p*-value | 0.0126 | 0.0023 | 0.1931 |
| Dinoflagellates (cell.L$^{-1}$) | sen.slope | 178 | 186 | 183 |
| | sen.slope.pct | 0.1329 | 0.1264 | 0.2407 |
| | *p*-value | 0 | 0 | 0 |
| *Phaeocystis* (cell.L$^{-1}$) | sen.slope | 25,767 | 247,220 | −7345 |
| | sen.slope.pct | 0.0854 | 0.3777 | −0.008 |
| | *p*-value | 0.4026 | 0.0109 | 0.922 |
| *Pseudo-nitzschia* (cell.L$^{-1}$) | sen.slope | 400 | 460 | 311 |
| | sen.slope.pct | 0.3385 | 0.4951 | 1.2563 |
| | *p*-value | 0 | 0 | 0 |

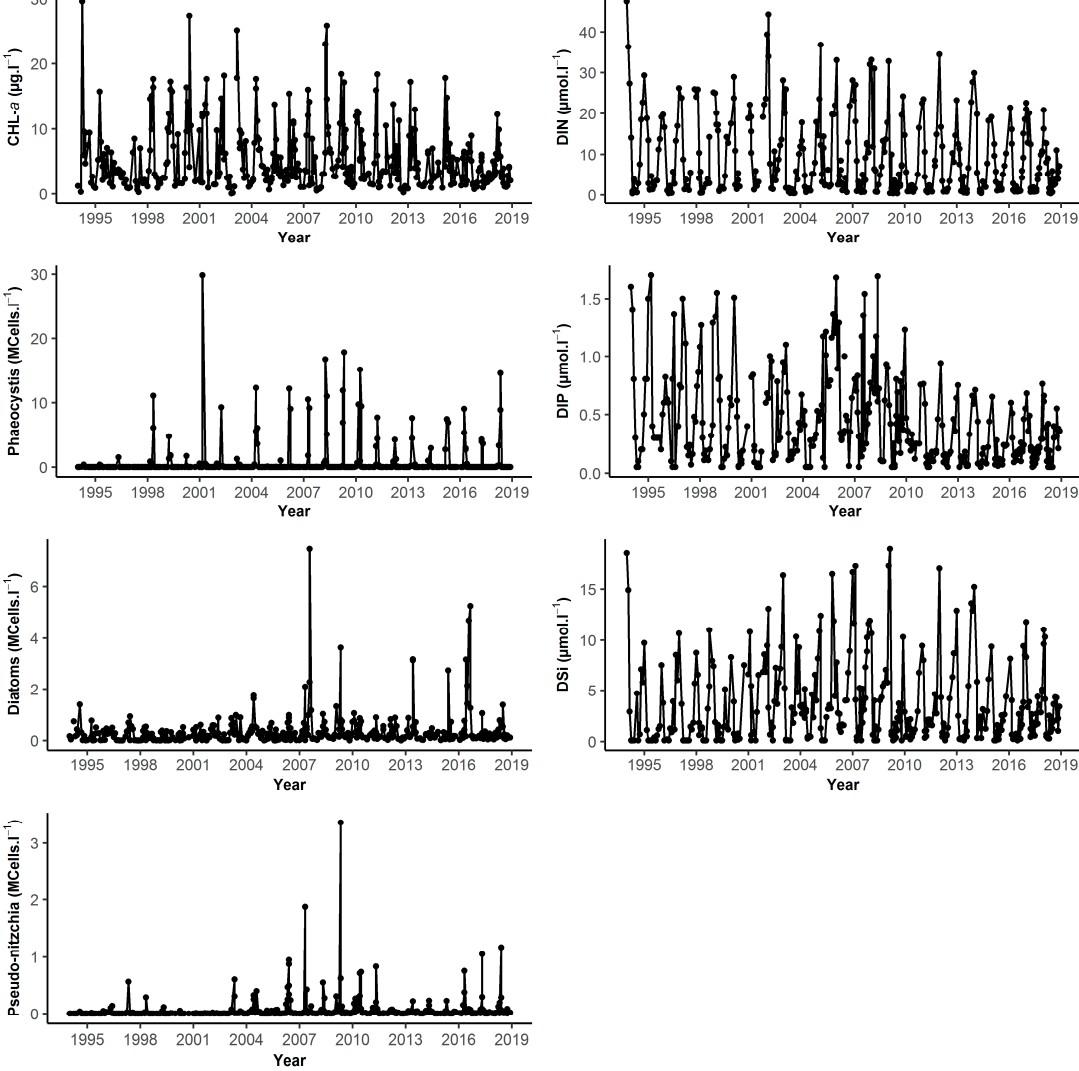

**Figure 3.** Annual time series of available parameters in the coastal station of Boulogne (BL1) for the period 1994–2018 (raw data).

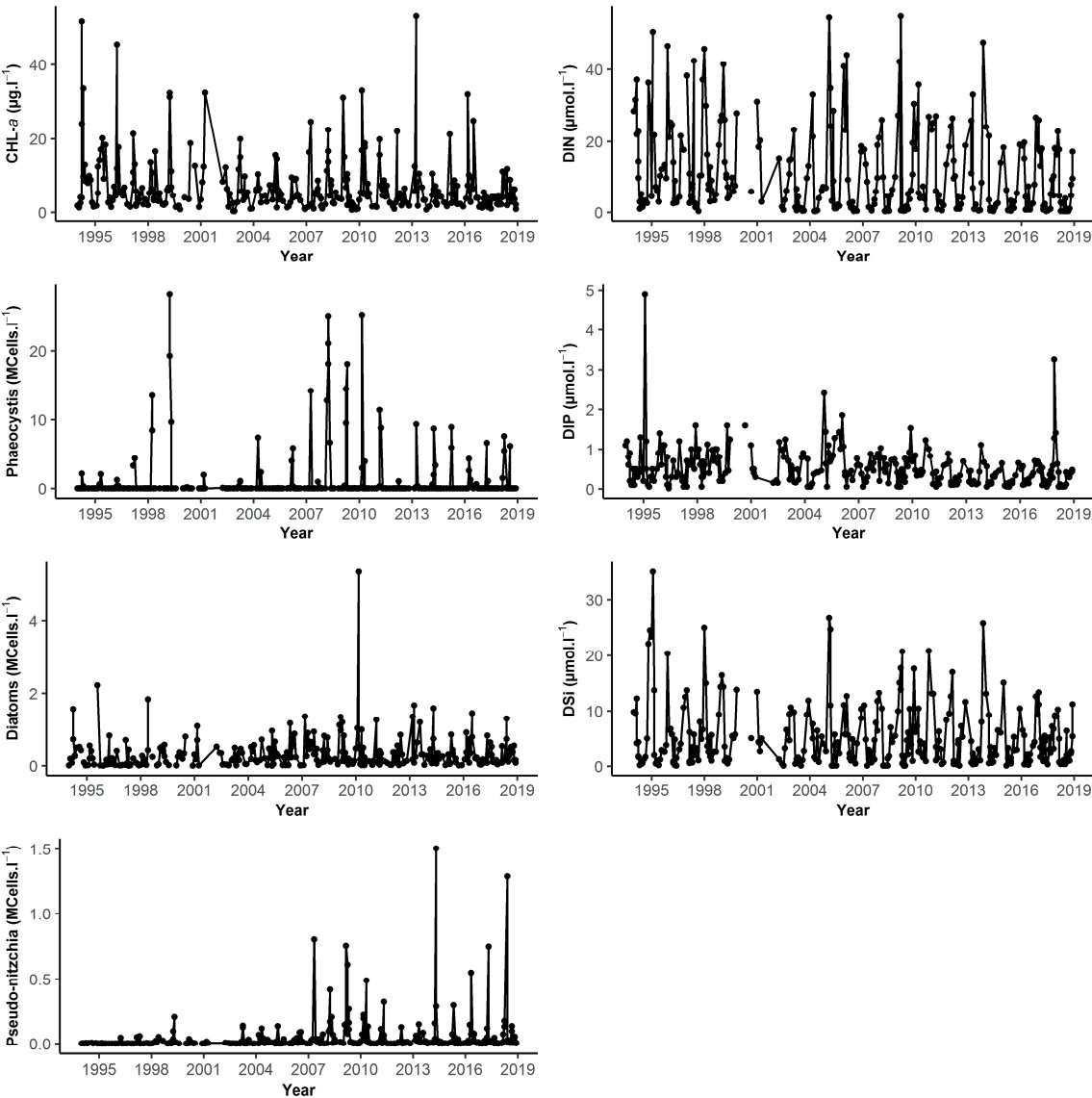

**Figure 4.** Annual time series of available parameters in the coastal station of Dunkerque (DK1) for the period 1994–2018 (raw data).

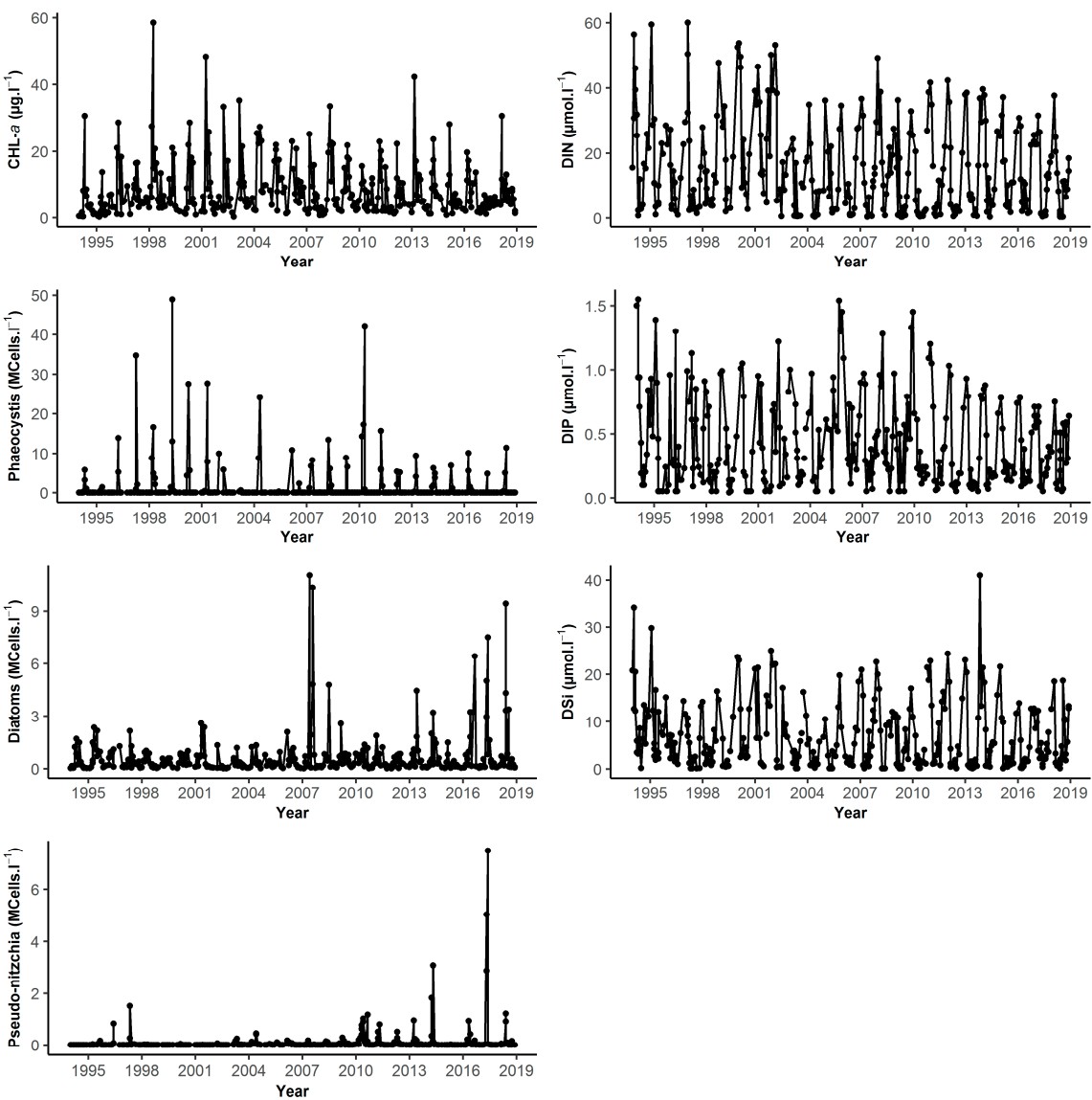

**Figure 5.** Annual time series of available parameters in the coastal station of the Bay of Somme (S1) for the period 1994–2018 (raw data).

## 3.2. Relationship between Phytoplankton Biomass, Abundance and Nutrients

Linear relations showed significant results between DIN and Chl-a, DIN and *Phaeocystis*, and DIN and diatom concentrations (Figure 6) using averaged maxima by studying the site for the whole period of interest 1994–2018. A detailed summary of model parameters and associated levels of significance is presented in Appendix B (Table A1). The assessed positive relationships are clear considering the intra-site coastal–offshore gradients but also when looking at the inter-site gradient, with a good overlap between sites. A clear gradient of ecosystem, from the most (Bay of Somme) to the least eutrophicated ecosystem (Boulogne-sur-mer), was highlighted (Figure 6).

When dividing the input dataset into several sliding sub-periods of six years, a clear positive linear relation was found between DIN and Chl-*a* for all periods and quite constantly in time (Figure 7, detailed plots per period are presented in Appendix C, Figure A2). The temporal trajectories of the relationship between DIN and Chl-*a* by site (Figure 8) confirm the validity of the general relationship previously established, with general individual trajectories nearly parallel to the average trend over the whole period of interest.

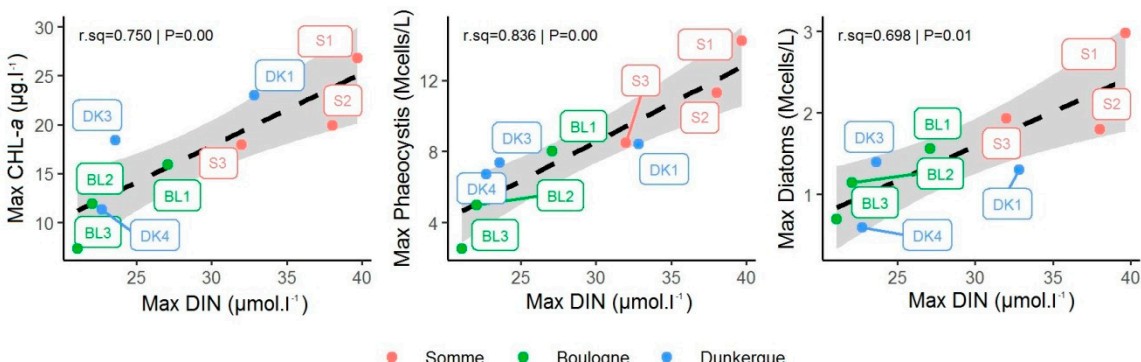

**Figure 6.** Relationships between Dissolved Inorganic Nitrogen (DIN) and Chlorophyll-*a* (Chl-*a*), DIN and *Phaeocystis globosa* and DIN and diatom concentrations by study site during 1994–2018, using averaged maxima for the whole period.

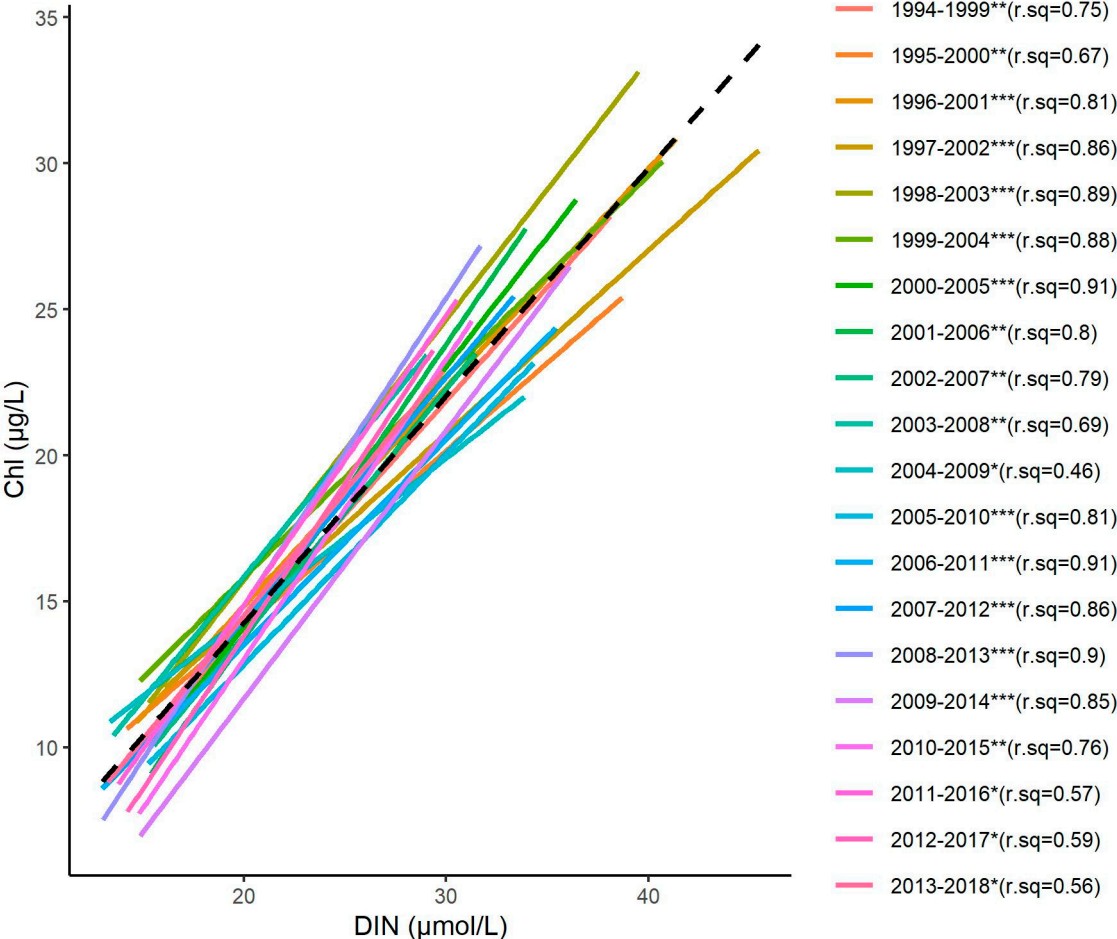

**Figure 7.** Linear regressions of the relationship between Chlorophyll-*a* (Chl-*a*) and Dissolved Inorganic Nitrogen (DIN) concentrations by six-year sliding period. Input values for both variables are averaged maxima by period, calculated during the growing season for Chl-a and during the winter season for DIN. *p*-values as well as r-squared are indicated in the legend, next to each corresponding period (0 '***' 0.001 '**' 0.01 '*' 0.05 '.' 0.1 ' ' 1).

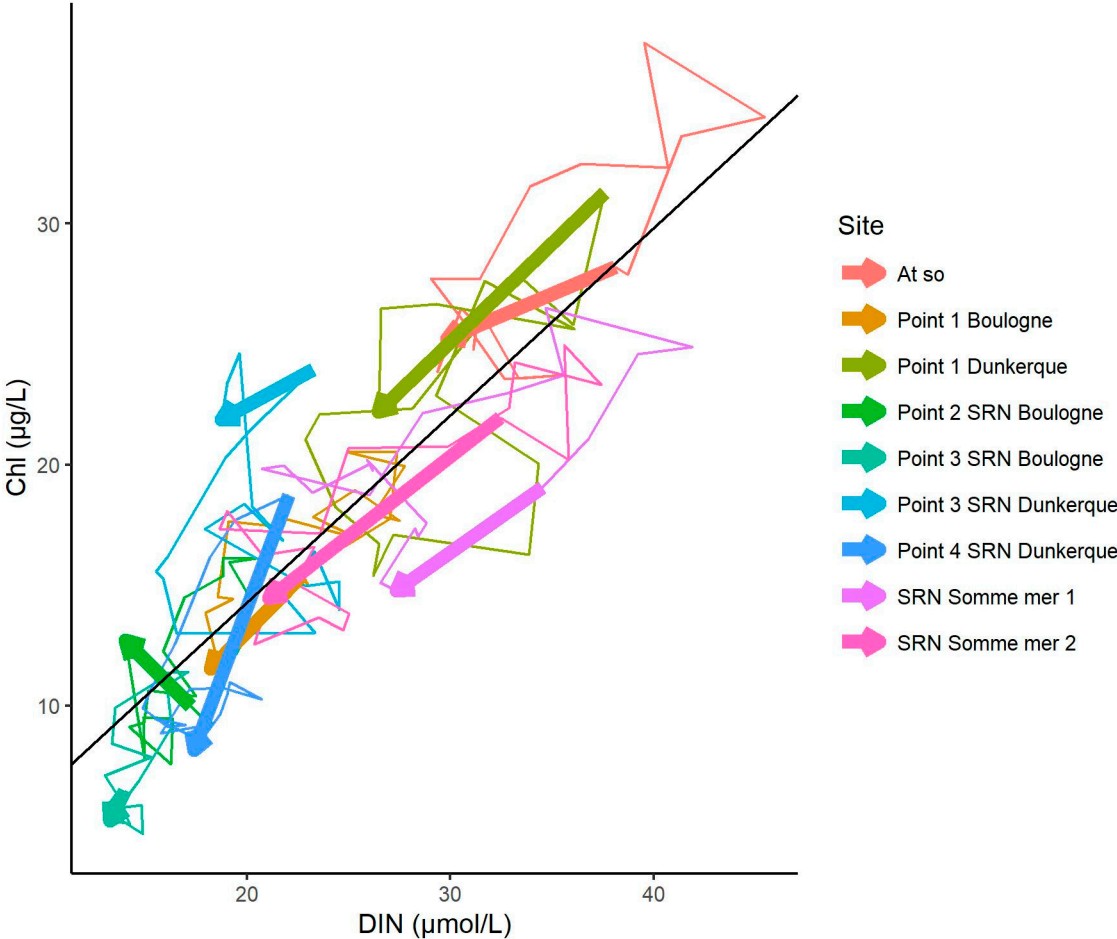

**Figure 8.** Temporal trajectories of the relationship between Chlorophyll-*a* (Chl-*a*) and Dissolved Inorganic Nitrogen (DIN) concentrations by site and by period (six-year sliding periods between 1994 and 2018). The arrows indicate the overall trajectories (first period to last period). Detailed trajectories are represented by thin colored lines.

### 3.3. Relationship between Phaeocystis and DIN Concentrations

The overall positive relationship between *Phaeocystis* and DIN concentrations by period shows contrasted results compared to Chl-*a*, with a decreasing trend in time (Figure 9; Detailed plots per period are presented in Appendix C, Figure A3). The shift from a steep to a moderate slope occurs around the years 2000 to 2008. Such a shift is due to the increase in *Phaeocystis* concentrations in sites presenting low initial *Phaeocystis* and DIN concentrations, while sites of highest initial *Phaeocystis* prevalence exhibit a decrease in its concentration, in coherence with the observed decrease in DIN concentrations (Figure 10).

### 3.4. Relationship between DIN and Diatom Concentrations

Regarding the relationship between diatoms and DIN concentrations, although it is positive during the whole period considered, its slope significantly increases over time, contrary to the case of *Phaeocystis* (Figure 11; Detailed plots per period are presented in Appendix C, Figure A4), with a shift at the beginning of the 2000s. This gives inter-site trajectories with very different directions from the general relationship previously established for the whole period of interest (Figure 12): Besides a decrease in DIN concentrations, diatom concentrations increase for all sites during the period considered. Such a pattern is very similar to what is observed among the diatom group for the genus *Pseudo-nitzchia* (Appendix D, Figures A5–A7).

### 3.5. Changes in the Competitive Advantage of Diatoms Versus Phaeocystis

The change in the relationship between the slopes of the regressions of *Phaeocystis* versus DIN concentrations on the one hand, and diatoms versus DIN concentrations on the other hand, indicates a shift from (1) a situation where the gradient in DIN concentrations was relatively more favorable to *Phaeocystis* growth compared to diatoms, to (2) a situation where the gradient in DIN concentrations was relatively more favorable to diatoms and less favorable to *Phaeocystis* (Figure 13).

### 3.6. Changes in DSi/DIN Ratio

The observation of the DSi/DIN ratio for all study sites indicates a statistically significant increase in time (Figure 14). A strong increase is observed from the period 1994–1999 to 2003–2008. Then, DSi/DIN slightly decreased but remained higher compared to initial values. This increase in the DSi/DIN ratio is more due to a DIN concentration decrease than a DSi concentration increase.

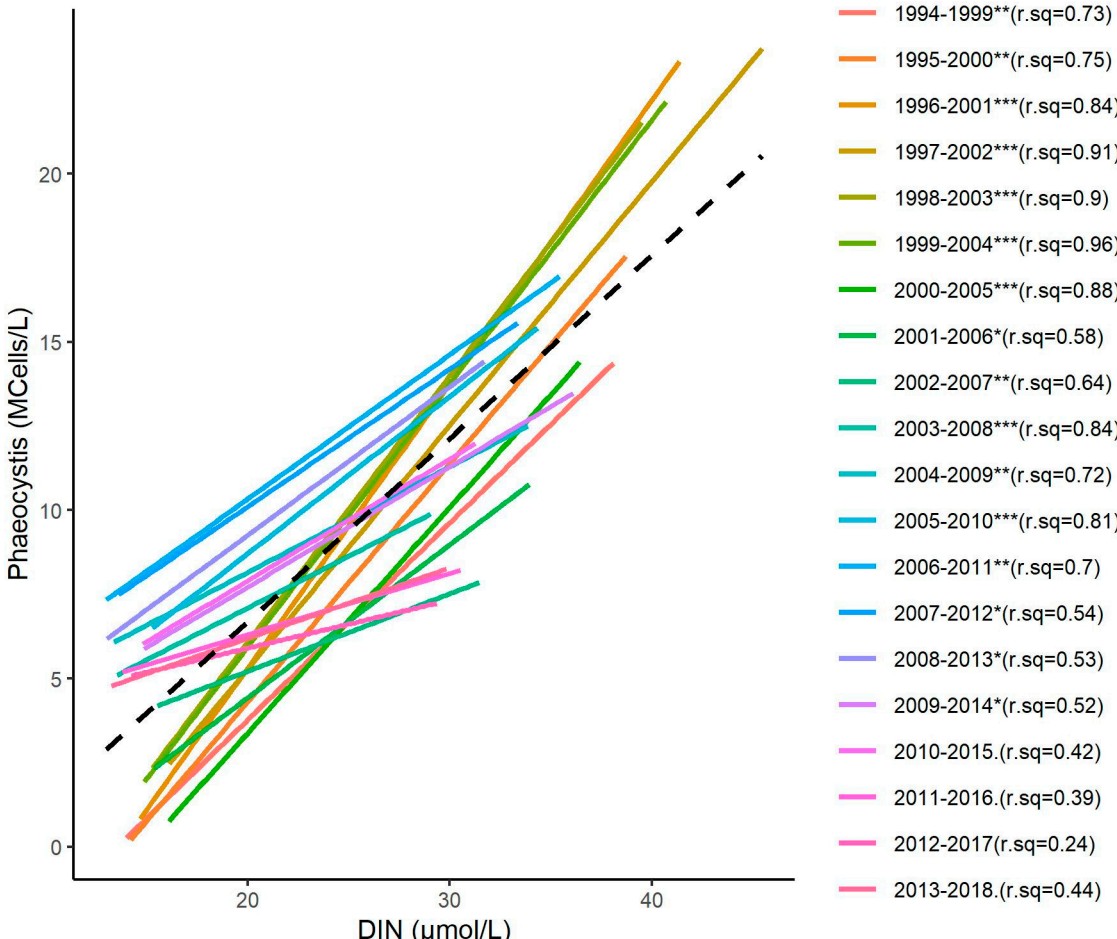

**Figure 9.** Linear regressions of the relationship between *Phaeocystis* and Dissolved Inorganic Nitrogen (DIN) concentrations by six-year sliding period. Input values for both variables are averaged maxima by period, calculated during the growing season for Phaeocystis and during the winter season for DIN. *p*-values as well as r-squared are indicated in the legend, next to each corresponding period (0 '\*\*\*' 0.001 '\*\*' 0.01 '\*' 0.05 '.' 0.1 ' ' 1).

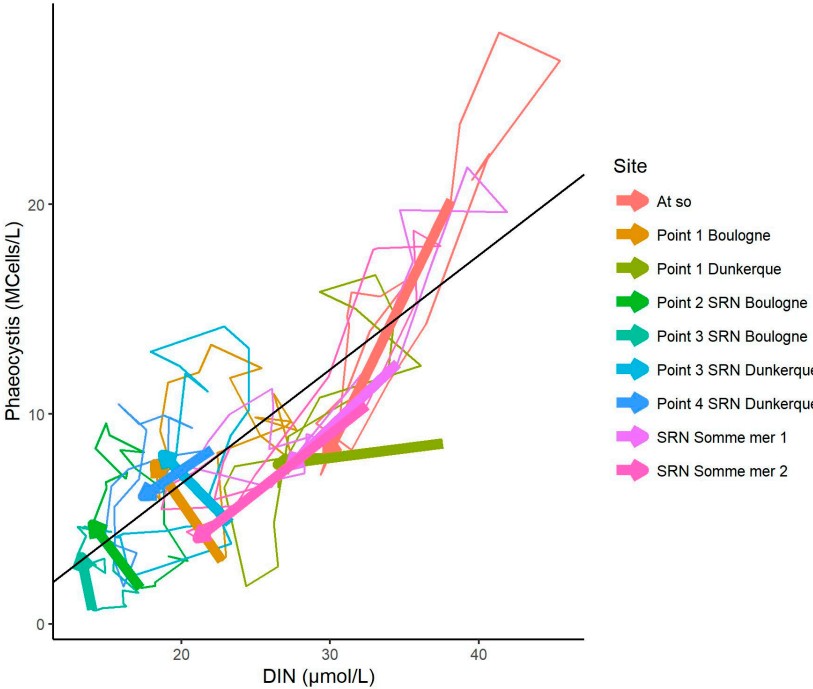

**Figure 10.** Temporal trajectories of the relationship between *Phaeocystis* and Dissolved Inorganic Nitrogen (DIN) concentrations by site and by period (six-year sliding periods between 1994 and 2018). The arrows indicate the overall trajectories (first period to last period). Complete trajectories are represented by thin colored lines.

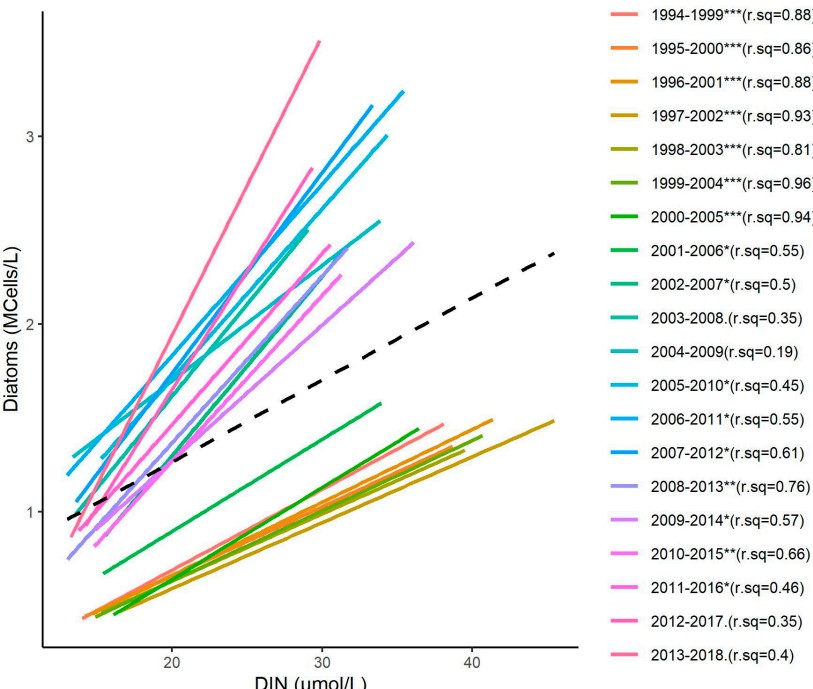

**Figure 11.** Linear regressions of the relationship between diatoms and Dissolved Inorganic Nitrogen (DIN) concentrations by six-year sliding period. Input values for both variables are averaged maxima by period, calculated during the growing season for diatoms and during the winter season for DIN. *p*-values as well as r-squared are indicated in the legend, next to each corresponding period (0 '***' 0.001 '**' 0.01 '*' 0.05 '.' 0.1 ' ' 1).

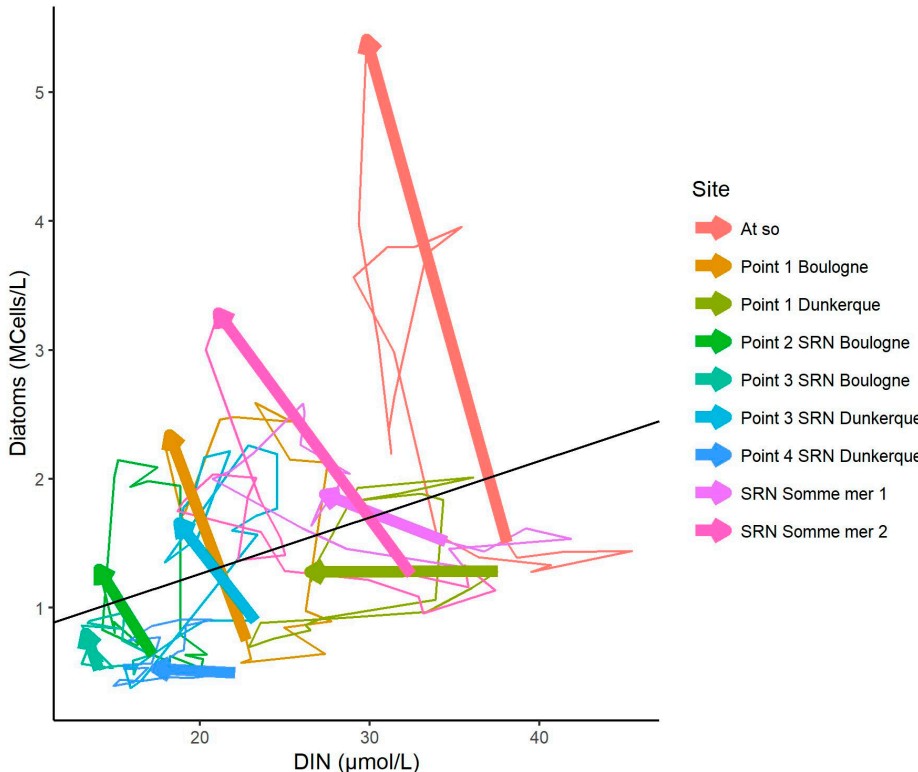

**Figure 12.** Temporal trajectories of the relationship between diatoms and Dissolved Inorganic Nitrogen (DIN) concentrations by site and by period (six-year sliding periods between 1994 and 2018). The arrows indicate the overall trajectories (first period to last period). Complete trajectories are represented by thin colored lines.

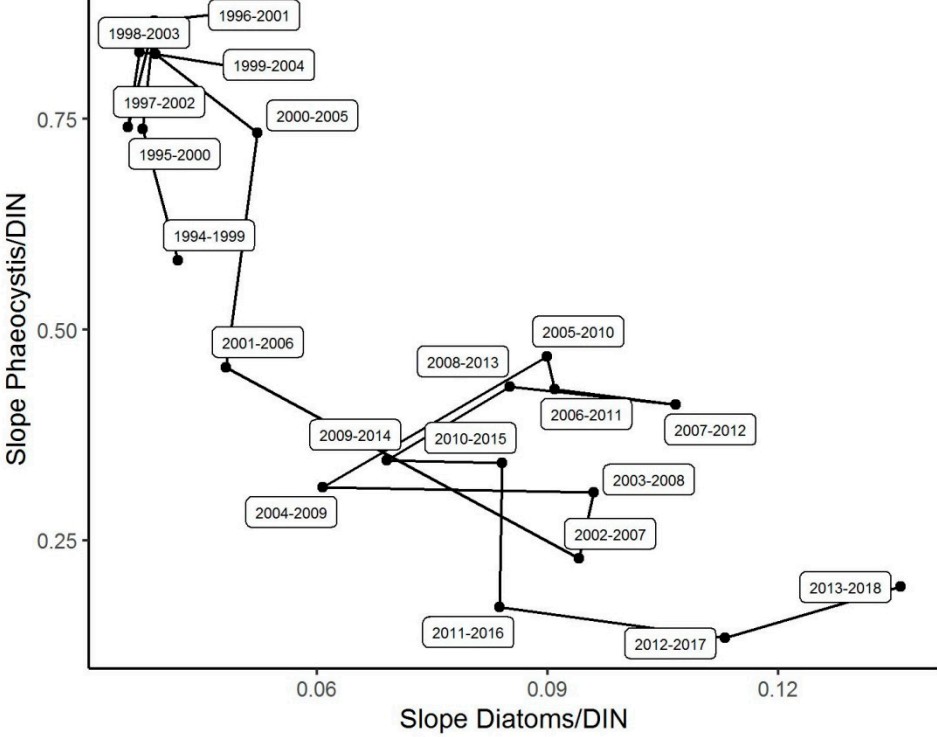

**Figure 13.** Temporal trajectory of the relationship between the slope of the *Phaeocystis* versus Dissolved Inorganic Nitrogen (DIN) and the diatom-versus-DIN regressions, by six-year sliding period.

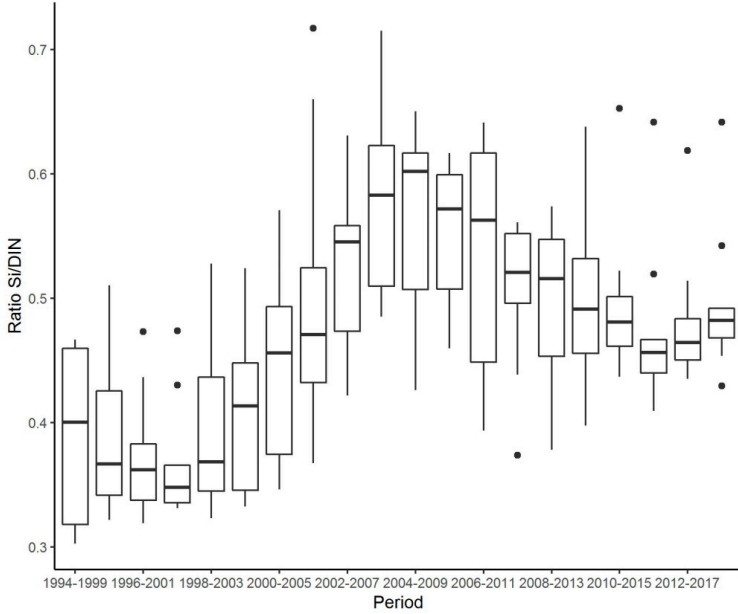

**Figure 14.** Boxplot of the Dissolved Silica (DSi)/Dissolved Inorganic Nitrogen (DIN) ratio by six-year sliding period, merged for all study sites. Input values for both variables are averaged maxima by period, calculated during the winter season for both DSi and DIN.

## 4. Discussion

Assessments of ecological or environmental quality status with regard to eutrophication, according to the WFD, the Marine Strategy Framework Directive (MSFD) [41] or the OSPAR Common Procedure, indicate that the eastern English Channel and the southern bight of the North Sea (for waters under French jurisdiction) are ecosystems in a rather moderate or poor status with a proven risk of eutrophication. Under-classifying factors are nutrient concentration and high phytoplankton biomass. The latter are clearly linked to the presence of the Prymnesiophyceae *Phaeocystis globosa*. As a result, this area has been considered for several decades by management measures to reduce nutrient inputs.

### 4.1. Long-Term Data Series, Research and Scientific Expertise and Advice

The SRN/REPHY/PHYTOBS data series constitute one of the longest time series at the French national level, allowing the study of the hydrology and phytoplankton compartments in the eastern English Channel and Southern Bay of the North Sea. The availability of such a time series covering the period 1994–2018 for a set of physico-chemical and biological parameters in contrasting environments of the eastern English Channel and the Southern Bay of the North Sea is of great interest in trying to answer a range of environmental questions. These questions may relate to the study of changes in phytoplankton dynamics, biodiversity and habitats in response to local anthropogenic forcing or more regional forcing related to climate change. Local to worldwide works have considered this long-term data series to study temporal changes [11,42,43], *Pseudo-nitzschia sp.* diversity [14], niche analysis [44] changes in diatom and dinoflagellate biomass (Belgian part of the North Sea) [45] and responses by phytoplankton to changes in precipitation [46]. Thematically, this kind of data series should improve knowledge of eutrophication processes and thus contribute to ecological and environmental status of assessments as advocated by the WFD and the MSFD.

Data users are informed that SRN/REPHY/PHYTOBS sampling stations are located along coastal-to-offshore transects under the local influence of anthropogenic pressures (from the more important in the bay of Somme to the less important one in Boulogne-sur-mer). Indeed, the geographical position of monitoring stations for studying eutrophication and phytoplankton dynamics is of primary importance when the geographical extent of river loads, and consequently impact of nutrient on

phytoplankton growth, can be highly variable according to local meteorological conditions driven by higher-level climatic influences [47].

There is a strong temptation for scientists to conduct a review by looking for the latest, best available numerical methodologies that have proven their power to analyze such a data set. Therefore, while discussion and communication of the results within the scientific community is possible and the results contribute significantly to the improvement of knowledge, it appears that matters become much more complex when it comes to addressing managers, policy makers, professionals and society at large.

Responses of phytoplankton to increasing, then decreasing anthropogenic pressures are sometimes less pronounced than expected because of resilience, hysteresis, and cumulative and/or synergetic effects. Patterns of change in the biomass and abundance of certain taxa (their pathway of change or trajectory) may appear complex, with no dominant direction when the time period of observation is short. On the other hand, as soon as the observation window is wide enough, a trajectory can emerge. This is what we will demonstrate with our results. To reach this goal, there are two main conditions: a need for high-quality time series available for the whole scientific community (to improve whole-plankton, multi-parameter approaches) and consideration of the long-term effects of management actions taken at time t. There is also a need, not to say an obligation, to get the message across to decision makers, stakeholders, environmental managers and society. This involves finding a compromise to communicate the best available science without over-simplification in the dialog about the existing complex interactions between processes, about irreversible change when a certain threshold is passed, about the dampening effects given a certain threshold, about change in the same or opposite direction as anthropogenic pressures change, and about long-term vs. short-term effects and consequences [19,48]. The more the numerical methodology used is complicated, the more the necessary popularization of the messages is complicated. Because of scientific integrity obligations, the scientist must discuss the limitations and uncertainties of methods and results. This can eventually backfire and lead to a lack of decision making on the pretext that nothing is certain. It is then sometimes difficult to reconcile political decision and scientific advice to avoid in creating frustration leading to inaction when the ecosystem does not react as expected following implementation of measures. There is a need to give the society a clear message and guidance (including targets ensuring the maintenance of key ecosystem functions and thereby ecosystem good and services) on how to improve environmental/ecological quality status.

In this context, a very simple, intuitive approach is proposed here to test the relationship between nutrient concentration and phytoplankton biomass and abundance, which is at the root of eutrophication problems. Based on this KISS (Keep It Smart and Simple) approach, and once the relationships are established for our ecosystems, it will be all the easier to mobilize policy makers and stakeholders to take remedial action. Obviously, this will not prevent the development of more complex, more sensitive and more complete numerical methodologies, but the results of these will have to be tested against this kind of easily understandable ground truth about the basic functioning of the ecosystems concerned.

When implementing our approach, the objectives were to test if (i) reducing nutrient riverine inputs has a rapid homogeneous positive effect on all studied ecosystems, (ii) the resulting patterns of change in the phytoplankton community structure favored non-HAB species and (iii) if the use of *Phaeocystis globosa* species indicator species is appropriate when dealing with eutrophication issues.

The pathways of change highlighted from our 25-year time series made it possible to identify different response patterns of the phytoplankton biomass and of the abundance of the Prymnesiophyceae *Phaeocystis globosa* and of diatoms (incl. *Pseudo-nitzschia sp.*) to nutrient riverine inputs management in three contrasted ecosystems in the eastern English Channel and the southern bight of the North Sea. Patterns of change are supposed to be representative of areas under temperate climatic conditions and they illustrate different ecosystems. From north to south: a shallow clear water area, transition from the English Channel to the North Sea ecosystem (waters offshore Dunkerque), a Region of Freshwater Influence (ROFI) (waters offshore Boulogne-sur-mer) and an estuary (the bay of Somme).

### 4.2. Relationships between Nutrient Concentrations and The Indicator of Phytoplankton Biomass

A clear, significantly positive relationship was observed between winter DIN concentrations and spring Chl-*a* concentrations, contrary to what was recently assessed by Desmit et al. [23] in the neighbouring Belgian coastal zone. This relationship is remarkably stable between periods, when using a synchronic approach. But what is striking is that when moving from a synchronic to a diachronic approach, by looking at individual intra-site trajectories in time, those trajectories are all nearly parallel to the mean trend observed, suggesting a very conservative relationship. This tends to indicate with quite low uncertainty that the three ecosystems studied are sensitive to nutrient input reductions. However, whereas the flow of macro-pollutants discharged by sewerage systems has been considerably reduced between 2000 and 2010 (Nitrate: 60%; Phosphate: 30%) at the whole watershed level (from Dunkerque to the bay of Somme), a clear decreasing trend only appears for phosphate, but not for nitrate, whose dynamics are more complex [49]. The ecosystem that shows the strongest evolution is in the Bay of the Somme with a clear reduction of phytoplankton biomass when nutrient concentration decreases. The Dunkerque site reacts in an intermediate way, while the Boulogne-sur-mer site shows the least marked change.

Trends observed for DIN and DSi are such that the DSi/DIN ratio increases until the early 2000s and then stabilizes or decreases slightly. At the end of the period, DIN inputs are more beneficial to diatoms than to *P. globosa*. DSi dynamics is rather different from site to site. Whereas no trend was identified for DK1, increasing and decreasing trends were observed, respectively, for BL1 and S1. This difference in DSi at sea can be explained by differences in water quality at the watershed level. Because of a more reduced riverine eutrophication near Boulogne-sur-mer (impacting BL1), DSi uptake by riverine diatom blooms is lower and, consequently, the export of silica to the coastal zone has increased in BL1 and this may potentially stimulate marine diatom production. Similar patterns were observed for coastal waters in the southern bight of the North Sea [50]. In the Bay of Somme, upstream DSi uptake by diatoms is still important because of a poor river water quality status. In this case, DIN inputs are more beneficial for *P. globosa*, while a DSi-limitation occurred for diatoms.

Results on biomass patterns are consistent with Gohin et al.'s [43] conclusion about a decreasing trend of phytoplankton biomass in the eastern English Channel in May, June and July correlated with lower river discharges and an historical (beginning of the 2000s) more efficient reduction of nutrient riverine inputs. Desmit et al. [23] also proposed such a conclusion for the North Sea with changes in phytoplankton phenology (onset and timing of the spring bloom occurring earlier in the year) in relation to de-eutrophication and sea surface warming with a tipping point around 2000. It is noteworthy that these authors did not show significant linear correlations between annual mean Chl-*a* and winter nutrient concentrations suggesting that complex interactions between factors such as nutrients, temperature, underwater light regime, and grazing may underlie the Chl-*a* trends.

### 4.3. Relationships between Nutrients Concentration and The Indicator Phytoplankton Species

Although increases in the abundance of *Phaeocystis sp.* in the Wadden Sea have been associated with eutrophication [51], North Sea populations have shown long-term fluctuations which appear unrelated to nutrient enrichment. In the Belgian coastal zone, nitrate enrichment has been identified as the principal factor regulating the magnitude of *Phaeocystis globosa* blooms and phytoplankton community [52]. Our results make it possible to consider with more confidence that the abundance of *P. globosa* is related to the concentration of DIN in the eastern English Channel (Bay of Somme, Boulogne-sur-mer transects) and the southern bight of the North Sea (Dunkerque transect). Nevertheless, the response of *P. globosa* to DIN varies depending on the initial state of the ecosystem under consideration. The pathways of change for *P. globosa*, diatoms and *Pseudo-nitzschia sp.* are more complex than those for Chl-*a*. At the beginning of the 2000s, a shift occurred in individual yearly trajectories compared to the overall trajectory defined on the period 1994–2018. *P. globosa* abundance increases in the historically less eutrophicated areas, whereas it decreases in the most eutrophicated ones. Diatoms and, more specifically, *Pseudo-nitzschia sp.* abundance increase for all areas. The evidence

of a shift in the early 2000s can also be linked to the findings from Alvarez-Fernandez et al. [53] with a decrease in dinoflagellates and an increase in diatoms annual maxima after 1998 in the North Sea. It is noteworthy that they also concluded an increasing proportion of dinoflagellates relative to diatoms (dinoflagellates not studied in the present manuscript). It seems that changing conditions push the environment above acceptable threshold levels, resulting in a system response when resistance to change was exceeded.

Considering changes highlighted in the eastern English Channel and southern bight of the North Sea, and under the hypothesis that these changes should be similar in other ecosystems in which *P. globosa* and *Pseudo-nitzschia* bloom, it is possible to forecast an intensification of HAB-related risks. Of course, the greatest risk is expected for estuarine and coastal waters where HABs occurrence are controlled by riverine nutrient inputs and also exacerbated by warming (most effective in shallow water) and the associated lower dissolved oxygen concentration and pH [54,55]. These pathways of changes for *P. globosa* and *Pseudo-nitzschia* should be considered as adaptation an (as defined by [48]) where "adaptation refers to the processes or coping strategies to be used by communities to increase their resilience (or decrease their vulnerability) to ecosystem changes". While actions to reduce nutrient inputs have been implemented for several years and are starting to show concrete results, the phytoplankton community seems to be adapting, thus modifying our perception of the changes that should occur.

### 4.4. Change Towards an Unstable Status

As demonstrated by Derolez et al. [29] for lagoons in the Mediterranean Sea, it seems that a trajectory with high variability (and feedback responses) suggests that the ecosystem has become unstable. The same instability is seen between years (even if a long-term pathway of change is identifiable) in our three ecosystems. It therefore appears that the implementation of nutrient input reduction measures is accompanied by a gradual return to a better state, but that this phase is particularly unstable. Our ecosystems are thus in an intermediate state between the initial state without pressure and the new, stable state after the application of management measures. In this unstable state, the ecosystem is very sensitive to any new (minor) modification of physical, biogeochemical or biological conditions. These new modifications could be from natural and anthropogenic origins (including climate change). According to Elliot et al. [48], as the nutrient pressures are removed, status may not improve for some time (Type I Hysteresis). It seems this stage has passed. However, as a complete resilience is neither expected nor impossible (the "return to Neverland" status from [26]), our ecosystems are gradually moving towards a lower status (Type II Hysteresis [48]). However, to reach this status, they inevitably go through this unstable phase. This phase could therefore be critical. Indeed, depending on whether efforts continue or not to reduce point source nutrient inputs, or depending on the addition of new pressures (e.g., chemical contaminants, climate change), this instability could cause the ecosystem to not continue its evolution towards the new stable state but to degrade again. Dynamics of this instability may also be altered by nutrient inputs from groundwater or from the atmosphere (diffuse, non-point sources).

The conclusion of Gomez and Souissi [56] about a generalized reduction of *P. globosa* in the eastern English Channel in response to de-eutrophication, mainly the reduction of river nutrient loads, is not fully supported by our study. However, their work draws our attention to the fact that, in these conditions, *P. globosa* is more vulnerable to anomalous climatic events (and maybe also extreme events: e.g., nutrient pulse in response to heavy rainfall) and maybe to competition with diatoms during the initiation of its theoretical bloom period.

This is to be linked to Karasiewicz et al. [44], who conclude that inter-annual variability in the magnitude and intensity of *P. globosa* and diatom blooms, compared to a decreasing trend in DIP and to less reduced DIN, rebalancing nutrient ratios suggest that other factors, such as competition for resources, may also play an important role. As highlighted by Duarte et al. [26], responses of coastal ecosystems to oligotrophication is more complex than expected and may be triggered by factors other

than nutrients, light and residence time. It is clear that the individual trajectories of the *P. globosa* and diatom groups are different. Shifts and subtle modifications occur in the relationship of each of these groups to nutrient intake but also between these groups via a certain competition, not reflected by the Chl-*a*/DIN relationship.

Of course, all other factor changes are not considered here and need our attention in future research and development of optimized monitoring programs (alteration of food webs, modification of ecological buffers, global changes, etc.).

For example, particular attention should be paid to benthic–pelagic nutrient coupling, which allows nutrients to be made available from sediment. Change toward a good ecological or environmental status should favor or mitigate this depending on bentho–pelagic coupling level. Derolez et al. [29] concluded that change toward a better status in Mediterranean lagoons needs a shift from pelagic-dominated to more benthic-dominated communities during the oligotrophication process. In the eastern English Channel, the pelagic–benthic coupling is stronger in shallow waters, such as in the stations studied [57,58]. This coupling will drive a shift of organic matter degradation to the benthic compartment, particle retention, and removal of nutrients from the biogeochemical cycle. Consequently, eutrophication studies and assessment should consider a strategy oriented toward a whole community approach, from epibenthic macrofauna, to phytoplankton and zooplankton, to large predatory fishes.

Improving knowledge about these complex trajectories and behaviors will enhance our capacity to forecast the restoration trajectories of ecosystems and to better define targets, ensuring maintenance of main ecosystem function to promote sustainable use of marine goods and services.

### 4.5. Limitations of Our Approach—Advice for Monitoring and Indicator Development

The advantage of our approach is to present clearly via a vector (i.e., the resulting trajectory over the period considered) and with a simple underlying numerical method the trajectories of some ecosystem components including multiple simultaneous pressures and changes considering random drifts and non-linear effects. The use of the average trajectory of biomass, indicator species abundance vs. nutrient (DIN) as a "reference" and observation of deviations from this reference is recommended in order to test the effectiveness of long-term management measures.

Whereas phytoplankton growth should theoretically be controlled by DIN:DIP:DSi ratios, our approach was mainly focused on DIN, since eutrophication status assessment in French marine waters does not considered DIP or DSi (no available threshold). The study of N:P ratio with our methodology is not possible. Indeed, the date for the maximum DIN value will very often be different from the date for the DIP value. Consequently, it is not possible to calculate the ratio from both maxima. Another option is to first calculate the N:P ratio and to consider the maximum. In this case, ratios can be quite extreme because DIP is highly fluctuating.

This kind of approach considering only phytoplankton biomass or indicator species such as *P. globosa*, *Pseudo-nitzschia* complex tends to oversimplify the assessment. The Chl-*a* concentration, used as a proxy of biomass, seems insufficient as a stand-alone indicator when taking into account the complexity of the phytoplankton response to changes in environmental pressures [30]. It is more or less the same when considering the diatom-to-dinoflagellate ratio, as it reflects the view that diatoms are good and flagellates or dinoflagellates are bad, which misunderstands the multiple roles that each group plays in the marine ecosystems [19]. Nevertheless, in the present study, we did not consider diatom changes only, as the potentially toxin-productive *Pseudo-nitzschia sp.* can also become dominant and modify community structure and function. This kind of KISS approach may be useful for many areas where there is a lack of consistent data to fully use a more holistic (at the community level) phytoplankton approach or to simplify the eutrophication assessment procedure when it is a question of carrying out assessment between several countries with limited resources for monitoring and analysis and, a desire for homogeneity of the results to be compared. Moreover, current metrics are partly based on this kind of ratio, e.g., PH1 Plankton Lifeform indicator [59], but we must not forget to include other indicators such as biomass and diversity in order to be able to truly claim to evaluate ecosystem health ([19]).

A lot of factors may cause time lags, regime shifts and non-linearity in ecosystem response following remediation efforts. The results show that, even several years after a major reduction in nutrient load, the objectives laid down by the WFD or RSC (OSPAR) have not fully been achieved in the eastern English Channel and the southern bight of the North Sea. As a result, the more recent targets set by the MSFD are also at high risk of not being met. Whereas it is important to provide clear recommendations based on easy-to-understand methodologies and results, there is a need for further research to better understand phytoplankton and HAB dynamics, i.e. the eutrophication/oligotrophication process, in general terms and to estimate the recovery times needed to reach a good ecological/environmental status. Such research should be based on integrated and sustainable coastal monitoring programs, including physics, biogeochemistry and biology, in a balanced way [60,61]. It is the responsibility of the scientific community to define the best available local-to-regional scale sustained monitoring programs and to develop early warning systems (using modelling and machine learning, e.g.,) to alert resource managers and stakeholders of HAB occurrences to take appropriate actions to reduce impacts.

**Author Contributions:** A.L. was particularly involved in data curation, funding acquisition, investigation, methodology, project administration, resources, supervision, validation and writing. C.D. mainly contributed to data curation, formal analysis, investigation, methodology, software and writing. All authors have read and agreed to the published version of the manuscript.

**Funding:** This work has been carried out through the project S3-EUROHAB (Sentinel-3 products for detecting EUtROphication and Harmful Algal Bloom events) funded by the European Regional Development Fund through the INTERREG France-Channel-England. Part of this work has been financially supported by (1) the European Union (ERDF), the French State, the French Region Hauts-de-France and Ifremer, in the framework of the project CPER MARCO 2015–2020; (2) The European Union's Horizon 2020 research and innovation program under grant agreement N° 654410 of the project JERICO-Next; (3) the Artois Picardie Water Agency (special thanks to J. Prygiel for his support).

**Acknowledgments:** The authors want also to express their gratitude to the Ifremer' LER/BL team: Devreker D. for graphic supports, Blondel C., Duquesne V., Hébert P., Cordier R. for technical support and Ifremer' VIGIES Department for data management. We sincerely thank the Sport Nautique Valericain, Aquamarine and the Haut de France Region's crew members of vessels involved in the collection of samples at sea.

**Conflicts of Interest:** The authors declare no conflict of interest. The funders had no role in the design of the study; in the collection, analyses, or interpretation of data; in the writing of the manuscript; or in the decision to publish the results.

## Appendix A

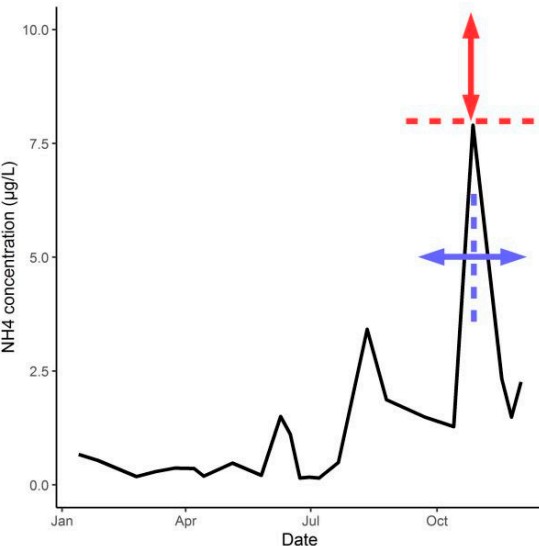

**Figure A1.** Illustration of the temporal (blue) and intensity (red) uncertainty of the "true" peak VS measured peak, here with the example of ammonium ($NH_4^+$).

## Appendix B

**Table A1.** Statistics of the regression lines for DIN~Chl-a, DIN~*Phaeocystis* and DIN~Diatoms (statistical significance of correlations are associated to a threshold alpha: significance is highlighted by * when *p*-value < alpha, 0 '***' 0.001 '**' 0.01 '*').

| Dependent Variable | Parameter | Mean [sd] | $R^2$ |
|---|---|---|---|
| Chl-a | Intercept | −3.05 [2.82] | 0.87 |
| | Slope | 0.85 [0.11] *** | |
| *Phaeocystis* | Intercept | −2.63 [1.72] | 0.84 |
| | Slope | 0.45 [0.07] *** | |
| Diatoms | Intercept | −0.51 [0.54] | 0.63 |
| | Slope | 0.08 [0.02] ** | |

## Appendix C

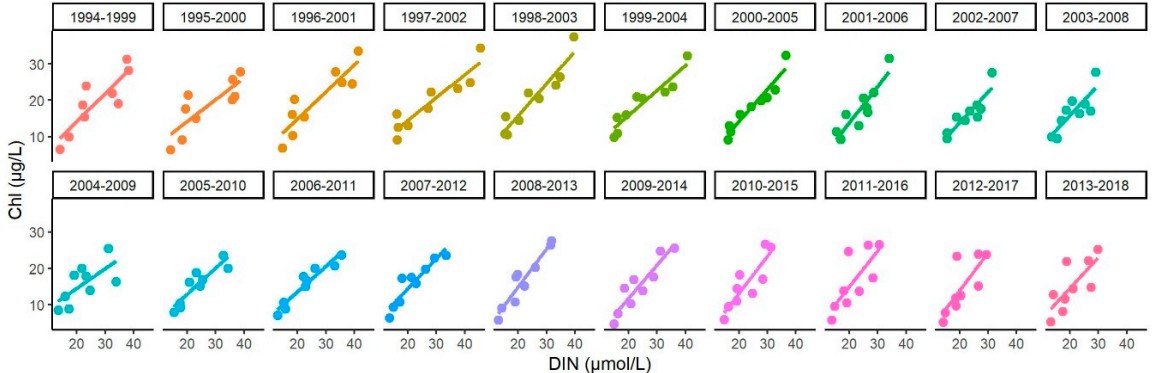

**Figure A2.** Linear regressions of the relationship between Chlorophyll-*a* (Chl-*a*) and Dissolved Inorganic Nitrogen (DIN) concentrations by period. Input values for both variables are averaged maxima by period, calculated during the growing season for Chl-a and during the winter season for DIN.

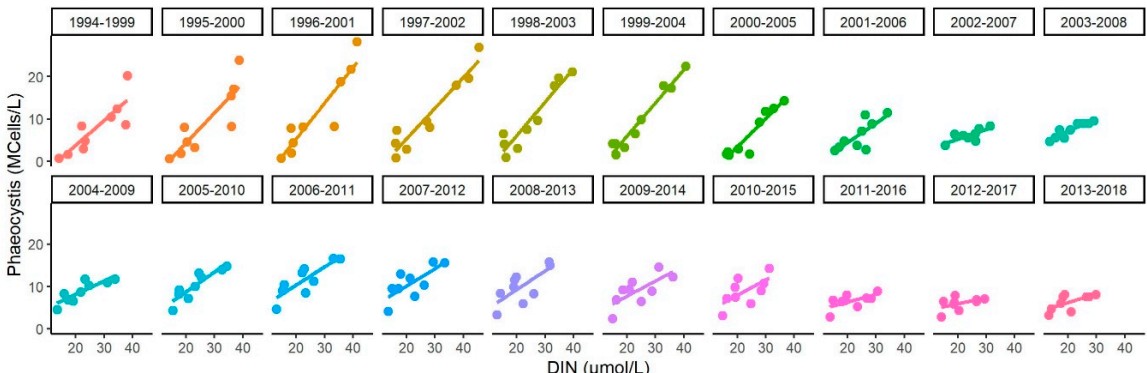

**Figure A3.** Linear regressions of the relationship between *Phaeocystis* and Dissolved Inorganic Nitrogen (DIN) concentrations by period. Input values for both variables are averaged maxima by period, calculated during the growing season for Phaeocystis and during the winter season for DIN.

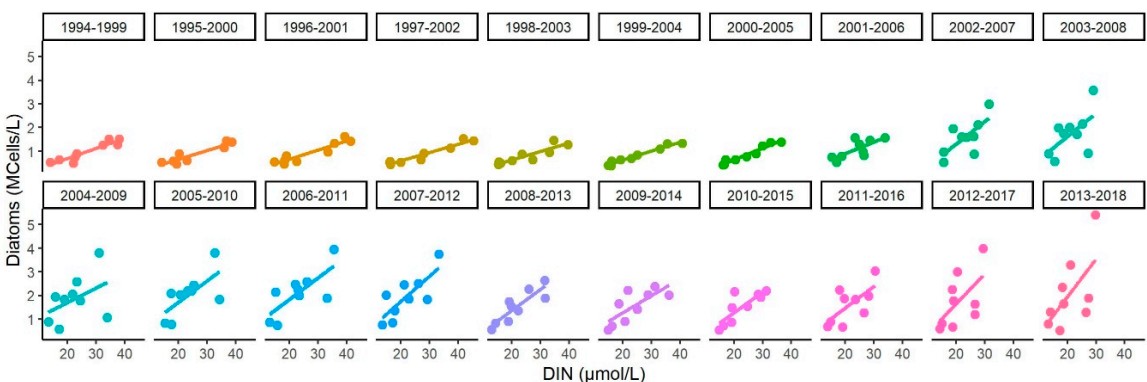

**Figure A4.** Linear regressions of the relationship between Diatoms and Dissolved Inorganic Nitrogen (DIN) concentrations by period. Input values for both variables are averaged maxima by period, calculated during the growing season for Diatoms and during the winter season for DIN.

## Appendix D

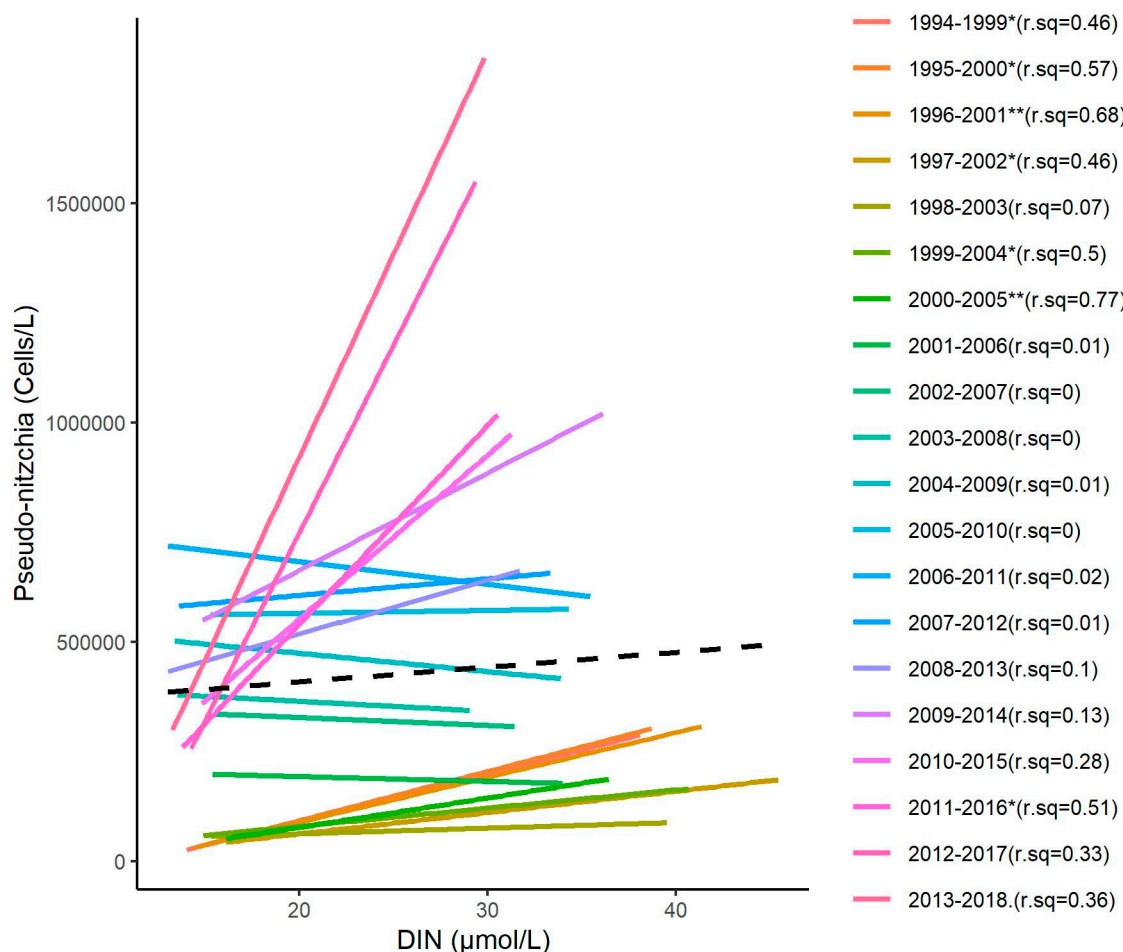

**Figure A5.** Linear regressions of the relationship between *Pseudo-nitzchia* and Dissolved Inorganic Nitrogen (DIN) concentrations by six-year sliding period. Input values for both variables are averaged maxima by period, calculated during the growing season for Pseudo-nitzchia and during the winter season for DIN. *p*-values as well as r-squared are indicated in the legend, next to each corresponding period (0 '***' 0.001 '**' 0.01 '*' 0.05 '.' 0.1 ' ' 1).

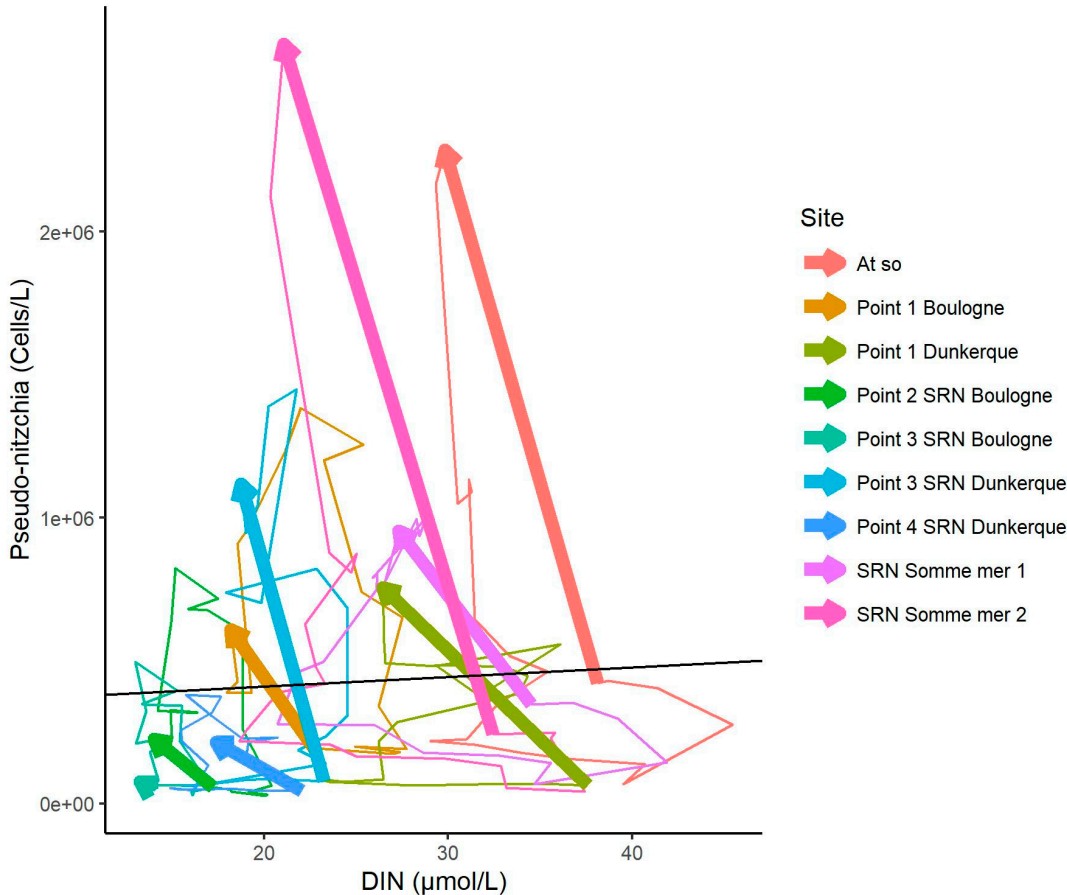

**Figure A6.** Temporal trajectories of the relationship between *Pseudo-nitzchia* and Dissolved Inorganic Nitrogen (DIN) concentrations by site and by period (six-year sliding periods between 1994 and 2018). The arrows indicate the overall trajectories (first period to last period). Complete trajectories are represented by thin colored lines.

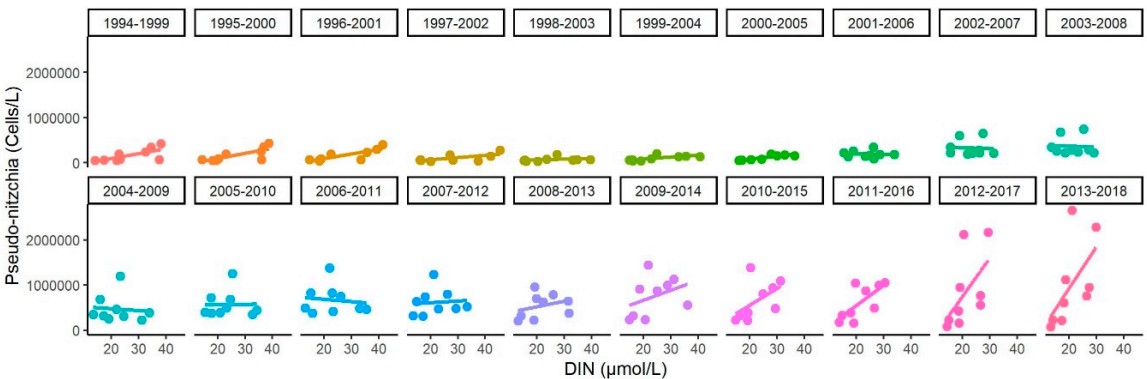

**Figure A7.** Linear regressions of the relationship between *Pseudo-nitzchia* and Dissolved Inorganic Nitrogen (DIN) concentrations by period. Input values for both variables are averaged maxima by period, calculated during the growing season for Diatoms and during the winter season for DIN.

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
