# Peer review of "Trajectories of Changes in Phytoplankton Biomass, Phaeocystis globosa and Diatom (incl. Pseudo-nitzschia sp.) Abundances Related to Nutrient Pressures in the Eastern English Channel, Southern North Sea"

_jmse, doi:10.3390/jmse8060401_

Round 1

Reviewer 1 Report

This manuscript makes an analysis of the changes in nutrient concentrations, chlorophyll and phytoplankton in the eastern Channel over a time period of 15 years. It deals with the question of the impacts on phytoplankon of nutrient enrichment and of the reduction in nutrient loads following measures to improve water quality and to achieve policy goals.

As the authors state, they have chosen “a very simple, intuitive approach” which may not include the most advanced statistical and scientific methods but makes it more easy to convey the message to policymakers and stakeholders. This is a very nice and elegant approach that helps to identify processes and responses in the marine environment related to the effects of “de-eutrophication”, that are often obscured by complex interactions. In particular, the response of phytoplankton biomass and composition to nutrient load reduction in the coastal waters of the Greater North Sea is not always unequivocal and leaves policymakers and environmental managers with many questions about the effectiveness of measures, and this approach makes it easier to communicate observations.

The authors show robust results that underline the relations between nutrient loads and phytoplankton, but there are still remaining questions about the underlying mechanisms that could be addressed in more detail in the paper, although it will certainly not be possible to solve all remaining issues:

  • the development in nutrient concentrations was studied, with the focus on DIN and DSi. DIP was not studied in detail (line 102-104) because of the importance of high DIN concentrations. But what about the possibility of changing ratios of N:P as a consequence of differences in the reduction of N and P loads, and the impact on the occurrence of either N or P limitation?
  • Similarly, the authors state that spring diatom blooms are controlled by DSi (line 78). But what about P-limitation, does this not occur in spring?
  • The relation between DIN and CHLa as shown with the 6-year moving averages in Figure 4 is very robust. It made me wonder if a similar result would be obtained if the annual values would have been used. Of course there will be slightly more variation due to interannual variability caused by factors like weather, irradiance etc?
  • What puzzled me is that Figure 5 shows a very clear, more or less unidirectional, response of CHLa to reductions in DIN. At the same time Phaeocystis (Fig 7) and diatoms (Fig 9) show a different response. Do the authors have an explanation for this apparent mismatch between chlorophyll and phytoplankton concentrations?
  • The authors show that the balance between diatoms and Phaeocystis has shifted and this coincides with a shift in DSi/DNI ratios. This can explain the decrease in Phaeocystis concentrations at the stations with high nutrient concentrations, but it does not explain why Phaocystis increased at the stations with low DIN concentrations (Fig. 7). Do the authors have an explanation for this remarkable phenomenon?
  • In line 412-413, the authors state “At the end of the period, DIN inputs are more beneficial to diatoms than to P. globosa”, but I would suggest that it is not the DIN input but the balance between DIN and DSi inputs that is beneficial for diatoms
  • Line 437: “Our results seem  to  prove  that  the  abundance  of    P. globosa  is  related  to  the concentration of DIN…”. In my view, ‘prove’ is a bit too strong as this is a correlative study and moreover, the response of Phaeocystis concentrations does not fully follow the reduction in DIN concentrations, as the authors also conclude in line 485.
  • Line 466-514: The authors discuss the shift of the ecosystem to a more ‘unstable’ state, but it is not clear to me how they define this unstable state
  • Figure A1: I think I understand what the authors want to illustrate with Figure A13, but the figure lacks explanation of what it exactly is that the authors want to tell.

Minor remarks:

  • The numbering of figures has gone wrong, there are two Figures 2, 3, 4
  • Figure 5, 7, 9: The colours for the different stations are a bit hard to distinguish. Is it an option to add a label to the various arrows to identify the different stations?
  • Line 427: sea surface warning should be sea surface warming

Author Response

Dear Reviewer,

first of all we want to thank you for your fruitfull comments and suggestion to improve our manuscript.

You will find attached the cover letter including our answers and all details of revisions that have been made.

Kind regards,

AL

Reviewer 2 Report

The presented manuscript will certainly be of interest to a wide range of readers. The authors conducted an original analysis of a large data array 1994-2018. But in the abstract, for some reason, 1992-2018 is indicated (line 15).
The text contains minor typos:
line 44 and 535 - possibly extra brackets
lines 198 and 200 - extra text "(data not shown)" "Error! Reference source not found". Obviously, you need to add a link to the literature.
Table 1 and 2 - you need to identify the separation of decimal places and the separator of groups of digits.
Line 551 - Supplementary Materials link www.mdpi.com/xxx/s1 does not work

Author Response

(The authors gave the same response as above.)
